Leaf arrangements are invalid in the taxonomy of orchid species

Jakubska-Busse Anna anna.jakubska-busse@uwr.edu.pl 1
Żołubak Elżbieta 1
Łobas Zbigniew 1
Gola Edyta Magdalena 2
1 Department of Botany, Institute of Environmental Biology, University of Wrocław , Wrocław , Poland
2 Department of Plant Developmental Biology, Institute of Experimental Biology, University of Wrocław , Wrocław , Poland
Escudero Marcial
Electronic publication date: 2017 Jul 21
Publication date: 2017
Volume: 5
Electronic Location ID: e3609
Received 2017 Apr 29; Accepted 2017 Jul 3
Copyright: ©2017 Jakubska-Busse et al.
Copyright year: 2017
Copyright holder: Jakubska-Busse et al.
License: This is an open access article distributed under the terms of the Creative Commons Attribution License, which permits unrestricted use, distribution, reproduction and adaptation in any medium and for any purpose provided that it is properly attributed. For attribution, the original author(s), title, publication source (PeerJ) and either DOI or URL of the article must be cited.
License URL: https://creativecommons.org/licenses/by/4.0/

Keywords: Epipactis, Taxonomy, Phyllotaxis, Orchids, Fibonacci pattern

Funding: University of Wrocław 1068/S/IBE/17 1076/S/IBŚ/2017 This work was supported by the University of Wrocław, grants Nos. 1068/S/IBE/17 and 1076/S/IBŚ/2017. The funders had no role in study design, data collection and analysis, decision to publish, or preparation of the manuscript.

==============================
The selection and validation of proper distinguishing characters are of crucial importance in taxonomic revisions. The modern classifications of orchids utilize the molecular tools, but still the selection and identification of the material used in these studies is for the most part related to general species morphology. One of the vegetative characters quoted in orchid manuals is leaf arrangement. However, phyllotactic diversity and ontogenetic changeability have not been analysed in detail in reference to particular taxonomic groups. Therefore, we evaluated the usefulness of leaf arrangements in the taxonomy of the genus Epipactis Zinn, 1757. Typical leaf arrangements in shoots of this genus are described as distichous or spiral. However, in the course of field research and screening of herbarium materials, we indisputably disproved the presence of distichous phyllotaxis in the species Epipactis purpurata Sm. and confirmed the spiral Fibonacci pattern as the dominant leaf arrangement. In addition, detailed analyses revealed the presence of atypical decussate phyllotaxis in this species, as well as demonstrated the ontogenetic formation of pseudowhorls. These findings confirm ontogenetic variability and plasticity in E. purpurata. Our results are discussed in the context of their significance in delimitations of complex taxa within the genus Epipactis.

Introduction

Understanding plant variability and the underlying genetic and developmental mechanisms are fundamental to modern plant classifications (Batista & Bianchetti, 2002; Jones & Clements, 2002; Rudall & Bateman, 2002; Bateman, Rudall & Moura, 2013). Genotypic and phenotypic variations reflect the adaptation of a plant to diverse and often demanding environments, and are generally accepted as driving forces behind speciation (Stace, 1991). The family Orchidaceae has recently been extensively studied in attempt to find the phylogenetic relationships within this family (Byng et al., 2016). Although some orchid taxa have been revised based on molecular markers (e.g.,  Tranchida-Lombardo et al., 2011; Bateman, Rudall & Moura, 2013; Fajardo, De Almeida Vieira & Molina, 2014; Zhao, Tang & Bi, 2017), there is still a lack of consensus regarding the delimitation of other genera Byng et al., 2016). Specifically, as yet there are no well-defined genetic markers for orchids which would enable the separation of e.g., aggregate taxa (Chung & Chung, 2012; Fajardo, De Almeida Vieira & Molina, 2014), especially regarding their phenotypic variability (Jakubska-Busse et al., 2017). Furthermore, the validation of the correct identification of plant materials for genetic analyses is mostly based on morphological traits. Therefore, taxonomic surveys focus mostly on flower and especially column (gynostemium) structure, acknowledged as the most reliable and stable characteristics in orchid classifications being related to the pollination systems (Mered’a, 1999; Szlachetko & Rutkowski, 2000; Gustafsson, Verola & Antonelli, 2010; Claessens & Kleynen, 2011; Jin et al., 2014). However, these surveys also include general morphological descriptions which are often used in manuals for the determination of taxa (Dressler, 1993; Szlachetko & Rutkowski, 2000; Delforge, 2006). One of these characteristics in taxa circumscription is leaf arrangement (e.g., Delforge, 2006); however, detailed data on this aspect in the Orchidaceae is lacking.

The phenomenon of regular and periodic patterning of leaves (or other lateral organs) is called phyllotaxis and has drawn the attention of researchers for centuries (e.g., Jean, 1994; Adler, Barabé & Jean, 1997; Reinhardt, 2005; Kuhlemeier, 2007). In the plant kingdom, two major types of leaf arrangements, whorled and spiral (helical) (Zagórska-Marek, 1985; Zagórska-Marek, 1994), are recognised. In whorled phyllotaxis, more than one leaf is simultaneously initiated at the meristem, forming a whorl of leaves in a node. The next whorl is circumferentially displaced so that its elements (leaves) are located in a mid-distance between leaves of the previous whorl. A special whorled leaf arrangement, called decussate phyllotaxis, occurs when two leaves are formed per whorl. This is a common pattern in, for example, the families Lamiaceae and Caryophyllaceae (Rutishauser, 1998; Reinhardt, 2005; Gola & Banasiak, 2016). Another modification of whorled phyllotaxis is distichy, whereby only one leaf is initiated per whorl, but the next leaf is displaced the half distance around the stem, i.e., 180°, with respect to the previous leaf. As a result, leaves occur in two opposite ranks along the stem. This leaf distribution is typical, for example, of the family Poaceae (Gola & Banasiak, 2016).

In the second major type of leaf arrangement, spiral phyllotaxis, successive leaves are initiated separately at the meristem and can be linked along the stem by a spiral line called the ontogenetic spiral. The spatial configuration (chirality) of the ontogenetic spiral can be either clockwise (S chirality) or counterclockwise (Z chirality) (Zagórska-Marek, 1985). Successive leaves are circumferentially displaced at a stable angular distance (divergence angle) and thus do not overlap (Zagórska-Marek, 1985; Jean, 1994). The most frequent spiral phyllotaxis in the plant kingdom (e.g., Zagórska-Marek, 1985; Zagórska-Marek, 1994; Jean, 1994; Adler, Barabé & Jean, 1997; Rutishauser, 1998) is related to the series of Fibonacci numbers, i.e., 1, 1, 2, 3, 5, 8, …etc., in which each element is the sum of the two preceding elements. The first number in this series refers to the ontogenetic spiral; however, this is hardly visible in the majority of shoots due to the shortening of internodes between successive leaves. Then the secondary spirals (parastichies), winding towards the apex in both directions, clockwise (S) and counterclockwise (Z), become visible at the stem surface. The most discernible spirals, crossing at right angles or near-right angles, form a contact parastichy pair, represented by the two succeeding numbers of the phyllotactic series, for example, 1:2 or 2:3 in the Fibonacci pattern (Adler, 1974; Zagórska-Marek, 1985; Zagórska-Marek, 1994; Jean, 1994). This formula unequivocally identifies the phyllotaxis of a given shoot (Zagórska-Marek, 1985; Zagórska-Marek, 1994).

It is generally accepted that the spiral leaf arrangement is a plesiomorphic feature in orchids, whereas distichous phyllotaxis or the presence of only one or two leaves per pseudobulb is an apomorphic trait (Withner, Nelson & Wejksnora, 1974; Arditti, 1992; Dressler, 1993). Rarely, due to the uneven elongation of internodes, two or more leaves are gathered at the same stem level (Dressler, 1993). In the genus Epipactis, the object of our analysis, leaves are usually reported as distichously arranged, but in some species spiral phyllotaxis can also occur (Dressler, 1993; Delforge, 2006; Brullo, D’Emerico & Pulvirenti, 2013; Byng et al., 2016, Lipovšek, Brinovec & Brinovec, 2017). Despite this general opinion, accurate data on phyllotaxis in the genus Epipactis suitable for use in manuals is lacking. This knowledge is however useful as the additional indirect confirmation of the species identity, especially during the field work when sometimes plants are available only in the vegetative phase and/or during revisions of the collections of plant specimens (vouchers). Therefore, the aims of our research were to (i) analyse the variability of leaf arrangements in E. purpurata in natural conditions; (ii) and quantify phyllotaxis diversity throughout the European range of the species based on herbarium materials, and thus (iii) to validate the usefulness of leaf arrangements in taxa identification.

Materials and Methods

Long-term field investigations of natural populations of E. purpurata were performed between 2003 and 2016 in permanent research plots in four locations in south-western Poland: in Nieszczyce near Rudna (51°32′14.26″N, 16°23′56.26″E), the “Błyszcz” nature reserve near Pątnów Legnicki (51°15′37.09″N, 16°12′56.95″E), Wałkowa near Milicz (51°30′00.46″N, 17°18′56.04″E) and Straża near Wińsko (51°23′51.40″N, 16°45′52.47″E). In this article, only the results of the analyses carried out in 2015 and 2016 are presented. In addition, relevant specimens from diverse geographical regions deposited in European herbaria (acronyms abbreviations after Thiers (2017): B, BR, C, FR, G, KTU, M, S, WRSL, Z, ZT), were analysed.

Leaf arrangements were analysed in both fresh and voucher specimens using the formula of a contact parastichy pair (Adler, 1974; Zagórska-Marek, 1985; Zagórska-Marek, 1994). In addition, a series of transverse sections through the mature vegetative shoots were prepared in order to indirectly confirm the leaf arrangements. At the moment when differences in leaf phyllotaxis became macroscopically visible, inflorescences had already been formed and shoot apical meristems were not available for detailed analyses.

For anatomical sectioning, shoot fragments which differed in leaf arrangements were collected and fixed in FAA (a formyl-acetic acid—50% ethanol mixture). Following dehydration in an increasing series of tertiary butyl alcohols (50%, 70%, 90%, 96%, and three changes in the pure butanol), the plant material was embedded in Paraplast X-tra (Sigma-Aldrich) and transversely cut, using a rotary microtome (Leica RM2135; Leica Instruments, Wetzlar, Germany), into 10–20 µm sections. Series of these cross sections were then de-waxed and stained with the Alcian blue-Safranin O mixture (1:1 v/v; O’Brien & McCully, 1981). Sequential digital images were taken using the system: a bright-field microscope Olympus BX 50— Olympus DP70 camera— Cell ˆ B software (Olympus Optical, Warszawa, Poland). Digital images were processed in Fireworks MX 2004 (Macromedia, San Francisco, CA, USA) and Photoshop CS6 (Adobe Systems, San Jose, CA, USA). Plant images were also taken in the field using Canon EOS 50D and Nikon D5300 cameras.

Experimental studies and material sampling were done with the permissions of the Regional Director for Environmental Protection, Nos.: WPN.6400.27.2015.IW.1., WPN.6205.122.2016. IL and WPN 6400.29.2016.IL

Results

In the course of our research, more than 470 ramets of E. purpurata were analysed in 2015 and 2016 in the field, along with over 800 individual herbarium specimens (Table 1). In the majority of shoots (1,210 shoots, i.e., 94.7% of all studied ramets), leaves were separately and spirally arranged along the stem (Figs. 1A and 2, Table 1). Their arrangement corresponded to 1:2 or 2:3 contact parastichy pairs, which are expressions of the main Fibonacci pattern. In the analysed material, the frequencies of both spatial configurations of spiral patterns were similar, with the ontogenetic spiral winding clockwise (S-chirality) in 51.9% and counterclockwise (Z-chirality) in 48.1% of cases.

Table 1 Leaf arrangements in the material analysed.

In a given shoot, more than one phyllotactic pattern can occur, as, for example, in ramets with a decussate pattern (see the text). In the table, for clarity, shoots with abnormal phyllotaxis (decussate pattern or with pseudowhorls) are counted only once within the total amount of ramets analysed. (A) Ramets from SW Poland: Nieszczyce (two cases) and the Błyszcz nature reserve (single case). (B) Phyllotaxis present in specimens from all herbaria analysed. (C) Herbarium voucher specimen details: Z–000088596; ZT–00071819. (D) Voucher specimens with aberrations in leaf arrangements and the acronyms of the herbaria collections are listed in Appendix S1.

		Leaf arrangement (phyllotaxis)	
		Spiral	Whorled decussate	Pseudowhorls	
	No. of ramets	No. of ramets	%	No. of ramets	%	No. of ramets	%	
Fresh material	477	450	94.4	3 A	0.6	24	5.0	
Herbarium vouchers	806	760 B	94.4	2 C	0.2	44 D	5.4	
Total	1,283	1,210	94.7	5	0.4	68	5.3	

In five cases (<1%), leaves were initiated in pairs (whorls) and oppositely inserted at the stem. Successive pairs were perpendicular to one another, forming a regular decussate pattern (Figs. 1B and 2, Table 1). In such shoots, two or, rarely, three whorls were present along the stem, while lower cauline leaves (below the decussate pattern) as well as bracts were arranged according to spiral phyllotaxis (Figs. 1B and 3). Interestingly, shoots with both decussate and spiral phyllotaxes were found in the ramets of one genet (Fig. 1B).

In several shoots (68 shoots, 5.3% of all analysed ramets), two or three leaves were gathered close to one another, seemingly at the same level of the stem (Figs. 1C, 1D and 3, Table 1). However, the leaves in such gatherings did not form opposite pairs and, in extreme cases, were distinctly inserted on one side of the stem (Fig. 3). The analysis of their spatial distribution proved that they were arranged according to the spiral Fibonacci pattern, which was continued along the whole shoot (Figs. 1D and 1E). Thus they were identified as pseudowhorls.

Figure 1 Diversity of leaf arrangements in E. purpurata.

(A) Typical spiral phyllotaxis; (B) An atypical decussate arrangement of E. purpurata shoots. Leaves are initiated in pairs (indicated by red arrows) which in successive nodes are perpendicular to one another. Note that the lower cauline leaves (indicated by yellow arrows) and bracts are inserted separately at the stem according to the spiral sequence, showing the ontogenetic transitions of the phyllotactic pattern; (C, D) Formation of pseudowhorls. Spirally initiated leaves gather seemingly at one level of the stem due to uneven internode elongation, forming pseudowhorls (indicated by red arrows). However, analysis of the leaf circumferential distribution proves the spiral sequence of leaf initiation (D). (E) Graphic representation of the leaf arrangement along the shoot presented in (C–D); red and blue lines represent parastichies winding toward the apex (black circle), i.e., from older to younger leaves in two opposite directions: clockwise (S chirality, blue lines) and counterclockwise (Z chirality, red lines); successive leaves are numbered, with 1 indicating the youngest leaf/bract and the highest number (13 or 14) indicating the oldest lower cauline leaf. Please note that the two ramets of a single genet presented in (C–D) are characterised by opposite chiralities of the ontogenetic spiral. (F) Developmental aberration in the shoot of E. purpurata. Two leaves differing greatly in size are visible at one level of the stem. Scale bars 5 cm (A–D) and 3 cm (F).

Figure 2 Graphic representation of leaf arrangements observed in E. purpurata shoots (drawn by Z. Łobas).

(A) Typical spiral distribution of leaves along the stem; (B) Decussate phyllotaxis; (C) Formation of pseudowhorls as a result of leaves gathering at the same stem level due to the limited growth (elongation) of the internode.

Figure 3 Voucher specimens of E. purpurata presenting a number of exemplary abnormalities in leaf arrangement and shoot development.

These abnormalities (indicated by filled black arrowheads) include pairs of opposite leaves (A, C, D, F) capable of forming a regular decussate pattern (C), pseudowhorls (B, E), extremely diversified sizes of leaves and split leaf tips (labelled with an asterisk, D), and a bifurcating shoot (F). Typically formed and arranged leaves below and above the nodes with decussate phyllotaxis or pseudowhorls are indicated by clear arrowheads. The acronyms of the herbaria and the voucher numbers are as follows: (A) DK-0005389, (B) DK-0005409, (C) Z-000088596, (D) ZT-00071775, (E) B 10 0591214, (F) FR-0001004. Scale bars 5 cm.

Histological analyses of mature shoots representing different phyllotactic patterns showed the arrangement of vascular tissue at the cross sections in relation to the leaf position. Vascular bundles were scattered throughout the cross section and distributed typically of monocotyledonous plants. In shoots with a spiral leaf arrangement, at one side of the stem, below the node, vascular bundles divided, giving rise to the leaf vasculature (leaf trace). These newly divided vascular bundles, at the level of leaf insertion in the node, diverged to the leaf, forming its supply system (Figs. 4B and 4C). After leaf departure, in the region of the internode, bundles were again relatively regularly scattered throughout the cross section. In the next node, the successive leaf trace was formed in the stem sector circumferentially distant ca. 137–140° from the previous leaf (Fig. 4B). This pattern repeated along the stem in relation to the successive leaves.

Figure 4 Histological analyses of the leaf vasculature in the shoots of E. purpurata differing in phyllotactic patterns.

(A–C) Diagrammatic representation (A) of the internode (B), node (C), and corresponding cross sections (B, C); dashed lines refer to the level of the cross section. Vascular bundles which will be incorporated into the leaf are already split in the internode (B) and visible in the cortical part of the stem. Later, in the node (C), they depart to form the leaf vasculature. (D–G) Vascular structure of a shoot with spiral phyllotaxis. (D) a graphic interpretation of the shoot, viewed from the top, shows the circumferential arrangement of three successively developed leaves (numbered 1, 2, and 3); the angle (circumferential displacement) between them is close to 137.5–140°and corresponds to the divergence angle for Fibonacci phyllotaxis. The cross section E–G present the same shoot at the levels corresponding to the nodes of three successively developed leaves (numbered 1–3). The positions of successive leaves are marked outside the cross sections as arcs. (H–K) Vascular structure of a shoot with decussate phyllotaxis. (H) a graphic interpretation of the shoot, viewed from the top, shows the circumferential arrangement of two pairs (numbered 1 and 2) of opposite leaves. Leaves of the second pair are circumferentially shifted and located halfway between those of the first pair; as a result, both pairs, the first and the second, are mutually perpendicular. Cross section (I–K) present the same shoot at the levels corresponding to the nodes of the first and the second pairs (numbered 1 and 2) of opposite leaves and the internode between them. The positions of successive leaf pairs are marked outside the cross sections as arcs. Scale bars (B–C, E–G, I–K) 1,000 µm.

In shoots with decussate phyllotaxis, the leaf trace formation for leaves of one pair occurred simultaneously at opposite sides of the stem (Fig. 4C). Vascular bundles in twofacing sectors split and departed, forming the vasculature of a given pair. In the subsequent node, the leaf traces for the next pair were again formed by the splitting of existing bundles, but in perpendicular sectors (Fig. 4C).

Discussion

In taxonomic descriptions of the species belonging to the genus Epipactis, the distichous and/or spiral leaf arrangement is usually cited as a typical pattern (Dressler, 1993; Delforge, 2006; Byng et al., 2016). However, during over a decade of research on Epipactis morphology, we found no distichous phyllotaxis. Importantly, it is contradictory to the data commonly used in manuals, where leaf arrangement—spiral or in two opposite rows (distichous), is even sometimes given as an indirect feature to distinguish between or to characterise separate species as well as complex taxa (aggregates), for example, E. atrorubens (Hoff.) Besser, E. tremolsii Pau and E. helleborine (L.) Crantz (Delforge, 2006). Interestingly, in 2016, we discovered an atypical decussate arrangement of leaves in E. purpurata. This prompted us to perform a detailed survey of phyllotaxis in this species. In the course of our research, using the formula of a contact parastichy pair, we indisputably disproved the presence of distichy in E. purpurata and confirmed the prevalence of spiral phyllotaxis as the typical leaf arrangement in this species. The phyllotaxis here was identified as that representing the most common Fibonacci pattern in plants (e.g., Jean, 1994; Adler, Barabé & Jean, 1997; Rutishauser, 1998). This pattern occurred in both spatial configurations with comparable frequency, indicating that the direction of the ontogenetic spiral in this species is randomly selected, similarly as in other plant species (Gregory & Romberger, 1972; Gómez-Campo, 1974; Zagórska-Marek, 1985; Zagórska-Marek, 1994).

The only exception to the typical spiral Fibonacci pattern in E. purpurata was the occurrence of the whorled decussate phyllotaxis. The decussate pattern in this species was established based on the circumferential arrangement of leaves and further confirmed by analysis of the vasculature. In monocot shoots, E. purpurata included (Jakubska-Busse et al., 2012), the vascular tissue forms a complicated network of bundles scattered throughout the cross section, which, in a longitudinal view, are inclined and wind spirally towards the apex (e.g., Pizzolato & Sundberg, 2002; Pizzolato, 2002; Pizzolato, 2004). Nevertheless, it is possible to establish the stem sectors in which the subsequent leaf traces are formed. Our results confirm that the arrangement of such stem sectors in E. purpurata was in accordance with the position of the leaf insertion at the stem, showing circumferential displacement in shoots with spiral patterns and a regular opposite arrangement in shoots with decussate phyllotaxis. Importantly, we have never observed the decussate pattern along the entire shoot; rather, it emerged during the development of a particular shoot, as leaves at its base, formed earlier in ontogeny, were separately initiated in a spiral sequence. Similarly, leaves above the decussate pattern, especially in the inflorescence, again represented Fibonacci phyllotaxis. These findings illustrate the repeated ontogenetic transitions between different patterns and indicate the developmental plasticity of the E. purpurata shoots.

Phyllotactic transitions are known to occur spontaneously during plant ontogeny along the same axis, and especially during the change of the developmental phase (Gómez-Campo, 1974; Meicenheimer, 1979; Meicenheimer, 1982; Battey & Lyndon, 1984; Zagórska-Marek, 1985; Zagórska-Marek, 1994; Kwiatkowska, 1995; Banasiak & Zagórska-Marek, 2006; Zagórska-Marek & Szpak, 2008), as well as being evoked by chemical factors (e.g., Maksymowych & Erickson, 1977; Meicenheimer, 1981). Among the immediate reasons for phyllotactic pattern transitions are variations in the geometric proportions between the organogenic zone of the meristem, where leaves are initiated, and leaf primordium size (Zagórska-Marek, 1987; Kwiatkowska, 1995; Zagórska-Marek & Szpak, 2008; Wiss & Zagórska-Marek, 2012). In meristems with the relatively wide organogenic zone and small primordia, various arrangements of primordia and thus different phyllotactic patterns are possible, as in magnolia gynoecia (Zagórska-Marek, 1994; Zagórska-Marek & Szpak, 2008; Wiss & Zagórska-Marek, 2012), cacti (Gola, 1997; Mauseth, 2004), or capitula of the Asteraceae (Hernandez & Palmer, 1988; Szymanowska-Pułka, 1994). In contrast, if primordia are relatively large compared to the organogenic zone of the meristem, only limited leaf arrangements are possible, as, for example, in grasses. Therefore, ontogenetic changes in apex geometry and the parameters of growth can affect primordia distribution and cause alterations in phyllotaxis. Interestingly, repeated changes in phyllotaxis due to altered meristem proportions have been proven so far only in two mutants, abphyl1 in maize (Jackson & Hake, 1999; Giulini, Wang & Jackson, 2004) and decussate in rice (Itoh et al., 2012). The increased diameter of the meristems of these mutants in response to an affected cytokinin signalling pathway causes a phyllotaxis transition upon development from the distichy in seedlings to the decussate pattern (Jackson & Hake, 1999; Giulini, Wang & Jackson, 2004; Itoh et al., 2012). A similar process is observed in Epipactis: early in ramet development, the spiral pattern is formed, then transformed during growth progression into a decussate leaf arrangement. However, in E. purpurata, this transition is unpredictable and occurs infrequently in populations (>1%). It is impossible to reach an indisputable conclusion about the developmental and/or genetic background of this phyllotactic change due to a lack of molecular tools for this species as well as to the rarity of the phenomenon and of the taxon itself. It can however be speculated that, similarly to abphyl1 and decussate mutants, developmental alterations in meristem size cause the observed phyllotaxis transitions. This can partially be confirmed by the fact that Epipactis shoots with decussate phyllotaxis always undergo a second transition back to the Fibonacci pattern during the change to the generative phase, during which the meristem size and growth parameters of the shoot are known to be significantly modified (e.g., Kwiatkowska, 2008). Additional evidence for the developmental plasticity of E. purpurata shoots is provided by the formation of pseudowhorls, i.e., gatherings of leaves seemingly located at one level of the stem due to the uneven elongation of internodes between them. Pseudowhorls are typical leaf arrangements in some species of Peperomia and Galium (Kwiatkowska, 1999; Rutishauser, 1999), and may also occur as a result of ontogenetic modifications of shoot growth, as in Anagallis (Kwiatkowska, 1995) and Acacia (Rutishauser, 1999).

Conclusions

In this article, we prove that in E. purpurata the spiral phyllotaxis is dominant; the presence of distichous leaf arrangement has not been confirmed. We document for the first time the presence of decussate phyllotaxis in E. purpurata, which is a rare exception to the typical spiral leaf arrangement in this species. We aim to draw the attention of orchid taxonomists to the intraspecific as well as the ontogenetic diversity of phyllotaxes in the Orchidaceae. Both phenomena are common in plants; moreover, even when one type of phyllotaxis prevails in a given taxon, it does not exclude the occurrence of other leaf arrangements; thus, the whole spectrum of possible phyllotaxes and their ontogenetic transitions must be considered in the course of taxonomic identification. Our finding of the atypical phyllotaxis is another example of ontogenetic variability in the genus Epipactis. In conclusion, we confirmed that spiral phyllotaxis is typical of E. purpurata and that the presence of other leaf arrangements has no taxonomic significance; this is another illustration of the developmental plasticity of the genus Epipactis.

Supplemental Information

Appendix S1 List of examined herbarium specimens of Epipactis purpurata, only with leaf aberrations

Click here for additional data file.

We thank to the Curators and Staff of the herbaria of B, BR, C, FR, G, KTU, M, S, WRSL, Z, ZT for loans of the specimens; and to the Reviewers for valuable comments on the manuscript.

Additional Information and Declarations

Competing Interests

Author Contributions

Field Study Permissions

Data Availability

The authors declare there are no competing interests.

Anna Jakubska-Busse and Edyta Magdalena Gola conceived and designed the experiments, performed the experiments, analyzed the data, contributed reagents/materials/analysis tools, wrote the paper, prepared figures and/or tables, reviewed drafts of the paper.

Elżbieta Żołubak and Zbigniew Łobas performed the experiments, analyzed the data, contributed reagents/materials/analysis tools, wrote the paper, prepared figures and/or tables, reviewed drafts of the paper.

The following information was supplied relating to field study approvals (i.e., approving body and any reference numbers):

Experimental studies and material sampling were done with the permissions of the Regional Director for Environmental Protection.

The following information was supplied regarding data availability:

The raw data has been supplied as Appendix S1.

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
