# Peer review of "Leaf arrangements are invalid in the taxonomy of orchid species"

_PeerJ, doi:10.7717/peerj.3609_

## Round 0.1 · original submission · Major Revisions

Dear Anna,

Please find attached the comments from two experts reviewers in taxonomy of orchids. They have made several suggestions that could improve the quality of your manuscript. Please, follow reviewers' recommendations and submit a new version of your manuscript and in particular please address the comments of Reviewer 1.

Sincerely,

Marcial.

Reviewer 1 ·

Basic reporting

In this study the authors examined a specific phenotypic trait (leaf arrangement) in an orchid species belonging to the taxonomic complex genus Epipactis. Specifically, authors, through a morphological survey of herbarium specimen and natural populations demonstrated the absence of the distichous phyllotaxy in this species. They also showed the presence of an atypical decussate phyllotaxis and the ontogenetic formation of pseudowhorls. The authors conclude that their data demonstrate the great ontogenetic variability and plasticity on E. purpurata.
The Manuscript is well written and data are well analysed and presented but I have major concerns about the premises and the relevance of the study.

Experimental design

Authors start from the consideration that the limited availability of molecular tools make it very important the identification of morphological traits for orchid classification. In this context I would have cited the paper from Tranchida et al. 2010 on molecular tools for species delimitation in Epipactis.
They also noticed that several traits as the column are highly indicated for this purpose because they can be in some way related to reproductive isolation and speciation. However, they focus on a trait that is in reality not very useful for species delimitation and that for sure has no evolutionary implications. Indeed, they found that it is inconsistent within a species.

Validity of the findings

Overall the authors showed that there is a great ontogenetic variability in the investigated traits. I fear that this finding was highly predictable and that results are not useful for the stated purpose of identifying distinguishing characters among orchid species.
Authors state that “the type of leaf arrangement is even used to distinguish between or to characterise complex taxa, for example, E. atrorubens, E. tremolsii and E. helleborine”. This is mainly not true basing on my sources as the three cited species are mainly defined on the basis of leaf traits other than their disposition. Also these species are even not universally accepted as species (see world monocot checklist). Even admitting that leaf arrangements would be an important trait for species categorization, they focus on a species E. purpurata that is generally identified basing on other traits.

Additional comments

In this study the authors examined a specific phenotypic trait (leaf arrangement) in an orchid species belonging to the taxonomic complex genus Epipactis. Authors claim that leaf arrangement is an important trait for species categorization but concluded that this trait cannot be used because it is inconsistent.
The Manuscript is well written and data are well analysed but I have major concerns about the premises and the relevance of the study.

Below are my main criticisms:

Authors start from the consideration that the limited availability of molecular tools make it very important the identification of morphological traits for orchid classification but they overlook studies as Tranchida et al. 2010. They also noticed that several traits as the column are highly indicated for this purpose because they can be in some way related to reproductive isolation and speciation. However, they focus on a trait that is in reality not very useful for species delimitation and that for sure has no evolutionary implications. Indeed, they found that it is highly variable within species.

Authors state that “The type of leaf arrangement is even used to distinguish between or to characterise complex taxa, for example, E. atrorubens, E. tremolsii and E. helleborine”. This is mainly not true and these species are even not universally accepted as species (see world monocot checklist) and are generally described using parameters other than leaf arrangements. Even admitting that leaf arrangements would be an important trait for species categorization, they focus on a species E. purpurata that is generally identified basing on different traits. In this context, their findings were widely predictable.

·

Basic reporting

There is some parts of the text that need additional references.

There is only one reference from the past four years, and not a lot more from this decade. I think that the authors should search for more recent literature to discuss to show that this kind of study still important.

The last figure should be colored to improve clarity.

The last goal of the study was not discussed enough.

Experimental design

A statistical test would corroborate the discussed results, because the numbers in the results seem very low to support the conclusions

Validity of the findings

The authors have background to discuss about the genus as a whole and they stated that it was one of the study goals.
Therefore, I think that this topic should be expanded in the discussion, improving the papers relevance.

A statistical test would corroborate the discussed results, because the numbers in the results seem very low to support the conclusions

Additional comments

The paper is interesting, but should be improved. A broader discussion about the genus is needed and the conclusions seem exaggerated.
Most of my suggestions are in the pdf file accompanying this review.

---

## Round 0.2 · accepted · Accept

Dear Anna,

I am glad we can access your ms for publication in PeerJ. Congratulations!

Sincerely,

Marcial.

Reviewer 1 ·

Basic reporting

This manuscript has been significantly improved in the present version.

Experimental design

The experimental design has been better justified in this revised version and is now clear

Validity of the findings

The manuscript has been improved by acknowledging all the criticisms made in the first revision and the findings are now placed in a wider context being thus more understandable.

Additional comments

Authors acknowledged all my comments on the previous version. I think data in this study are well presented and analysed and are placed in the wider context of the literature on this topic.